# Shorter cortical adaptation in dyslexia is broadly distributed in the superior temporal lobe and includes the primary auditory cortex

Sagi Jaffe-Dax[1]*, Eva Kimel[2], Merav Ahissar[2,3]

[1]Department of Psychology, Princeton University, Princeton, United States; [2]The Edmond and Lily Safra Center for Brain Sciences, Hebrew University of Jerusalem, Jerusalem, Israel; [3]Department of Psychology, Hebrew University of Jerusalem, Jerusalem, Israel

**Abstract** Studies of the performance of individuals with dyslexia in perceptual tasks suggest that their implicit inference of sound statistics is impaired. Previously, using two-tone frequency discrimination, we found that the effect of previous trials' frequencies on the judgments of individuals with dyslexia decays faster than the effect on controls' judgments, and that the adaptation (decrease of neural response to repeated stimuli) of their ERP responses to tones is shorter (*Jaffe-Dax et al., 2017*). Here, we show the cortical distribution of these abnormal dynamics of adaptation using fast-acquisition fMRI. We find that faster decay of adaptation in dyslexia is widespread, although the most significant effects are found in the left superior temporal lobe, including the auditory cortex. This broad distribution suggests that the faster decay of implicit memory of individuals with dyslexia is a general characteristic of their cortical dynamics, which also affects sensory cortices.
DOI: https://doi.org/10.7554/eLife.30018.001

**\*For correspondence:**
jaffedax@princeton.edu

**Competing interests:** The authors declare that no competing interests exist.

## Introduction

Dyslexia, a specific and significant impairment in the development of reading skills that is not accounted for by mental age, visual acuity problems, or inadequate schooling (*World Health Organization, 2016*), affects ~5% of the world's population (*Lindgren et al., 1985*). Though individuals with dyslexia are diagnosed for their reading impairments, they also often have difficulties in simple non-linguistic perceptual tasks (*Mcanally and Stein, 1996*; *Ahissar et al., 2000*; *Sperling et al., 2005*; *Giraud and Ramus, 2013*). These can be largely explained as resulting from inefficient use of stimulus statistics that characterize the experiment (the 'Anchoring Deficit hypothesis'; *Ahissar et al., 2006*; *Oganian and Ahissar, 2012*; *Jaffe-Dax et al., 2015*). In these tasks, participants are not aware of the effect of previous stimuli. But their perception — in particular when retention is required, as for example in serial discrimination tasks — tends to contract to the estimated mean of previous stimuli (contraction bias; *Raviv et al., 2012, 2014*). This contraction to the mean merges the (implicit) predicted stimulus (based on previous exposures) with the current sensory estimate, forming a coherent percept.

The neural mechanism that may underlie the implicit learning of experimental statistics is adaptation; that is, an automatic, implicit, and stimulus-specific decrease of the response to repeated stimuli. Importantly, the rate of decay of the behavioral effect of previous trials in serial discrimination is similar to the rate of decay of neural adaptation, as measured by magnetoencephalography (MEG) (*Lu et al., 1992*). Inspired by this finding, we recently compared both the behavioral dynamics

**Figure 1.** Cortical distribution of the groups' mean estimated time constants (τ) of adaptation, calculated separately for each of the responding voxels. (**A**) Control participants. (**B**) Participants with dyslexia. The estimated τs for participants with dyslexia were consistently shorter than those estimated for the control group. Significant group differences in the whole-brain analysis (Monte-Carlo cluster-level corrected: cluster threshold of 44 voxels; see 'Materials and methods') are outlined in magenta. The left and right primary auditory cortices, which were estimated as a source of P2 (ERP) component, are outlined in orange. An ROI analysis (see text) revealed a significant group difference in the left primary auditory cortex (***Figure 2***).
DOI: https://doi.org/10.7554/eLife.30018.002

and the rate of adaptation (event-related potential [ERP] responses) of good readers (i.e., the control group) and dyslexic participants (***Jaffe-Dax et al., 2017***). All participants performed serial two-tone frequency discrimination in four blocks with different Trial Onset Asynchronies (TOAs, i.e., temporal interval from the onset of a trial to the onset of the next trial). Both the magnitude of perceptual contraction to the mean frequency of previous trials and the magnitude of neural adaptation (P2 and N1 components that are automatically produced by the auditory cortex [***Mayhew et al., 2010***]) decayed faster in participants with dyslexia (ERP [***Jaffe-Dax et al., 2017***]).

As ERP responses cannot be used to localize the cortical source of this group difference, we now recruited the participants from the ERP study (***Jaffe-Dax et al., 2017***) to take part in an fMRI study with a similar protocol, which allowed us to characterize which brain areas show shorter adaptation in dyslexia. Using the ERP-based protocol in the scanner, we measured the BOLD response ($\beta$s) to tones for each TOA, and calculated the time constant of adaptation (fitting an exponential decay function). To identify the areas of significant group difference in time constants of adaptation, we used two methods: data-driven whole-brain analysis, and hypothesis-driven regions of interest (ROI) based on areas suggested to produce N1 and P2 (e.g. ***Mayhew et al., 2010***). All cortical regions that responded to tone discrimination showed a tendency to decay faster in participants with dyslexia. Whole-brain analysis revealed significant group differences in the left superior temporal lobe and in the right insular cortex, and ROI analysis revealed significant differences in the left primary auditory cortex.

## Results

We recruited 20 participants with dyslexia and 19 good readers from our previous study (***Table 1***; ***Jaffe-Dax et al., 2017***) and asked them to perform two-tone frequency discrimination in separate

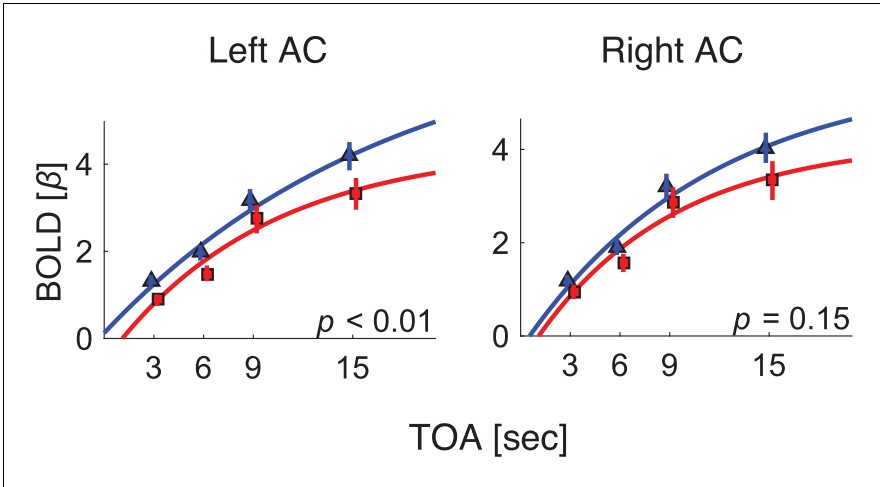

**Figure 2.** BOLD response as a function of TOA in the primary auditory cortex of each hemisphere.  Blue: control. Red: dyslexic. AC: the 3 subregions that comprise the primary auditory cortex, outlined in orange in *Figure 1*.
DOI: https://doi.org/10.7554/eLife.30018.003

blocks with four trial-onset intervals (TOAs) of 3, 6, 9, and 15 s. Before entering the scanner, all participants performed a short four-block training session with simulated scanner noise to familiarize them with in-scan conditions (*Sperling et al., 2005*; *Chait et al., 2007*). The two groups exhibited similar accuracy (72.4 ± 6% vs. 73 ± 4.6%, $z = 0.5$, $p=0.57$). In scans, good readers (controls) performed better than dyslexic participants (Mean ± SEM: 82.5 ± 1.6% vs. 76.3 ± 2.2%, $z = 2.6$, $p<0.01$ in Mann-Whitney U-tests), suggesting that they gained more from the short pre-scan practice (in line with the faster learning reported by *Jaffe-Dax et al., 2017*).

To evaluate the dynamics of cortical adaptation in each group, we used the following procedure. First, we determined which Talairach voxels responded to the task (standard generalized linear model [GLM], $p<0.001$, false discovery rate [FDR] corrected) when all participants were considered. For each of these voxels, we calculated the dynamics of adaptation, among control participants and among participants with dyslexia, as follows. We estimated $\beta$ over the mean blood oxygenation level dependent signal (BOLD) response of each group in each of the four TOA conditions (block design; see 'Materials and methods'). Using these $\beta$s, we fitted an exponential decay model (*Lu et al., 1992*; *Jaffe-Dax et al., 2017*): $\beta(TOA) = a + b\exp(-TOA/\tau)$ to each voxel. In this model, $\tau$ denotes the time constant of adaptation, $a$ is the asymptotic level of BOLD and $b$ is the magnitude of adaptation. *Figures 1A and 1B* show the distribution of the mean fitted $\tau$s for the control and dyslexic groups, respectively. Their comparison illustrates the broadly distributed trend of faster decay in the dyslexic group.

To locate regions in which the fitted $\tau$ differs significantly between the groups, we conducted a whole-brain analysis, in which we fitted $\tau$ to each voxel, for each participant separately. To reduce the impact of outliers resulting from the noisy estimation of $\tau$ (due to this single subject and single voxel analysis), we assessed group difference with a non-parametric test (Mann-Whitney U test), in which extreme values are not over-weighted. We corrected for multiple comparison bias by requiring a cluster of contingent voxels with a significant group difference (cluster corrected for $p<0.05$, dictated 44 spatially contingent voxels, based on Monte-Carlo cluster-level correction). Significant regions were found in the left superior temporal cortex (TAL: −54,–18, 10) and in the right insular cortex (TAL: 39,–2, −8), outlined in magenta in *Figure 1A and 1B*. The superior temporal cortex is known to be involved in a broad range of auditory tasks, including simple tone discrimination (*Daikhin and Ahissar, 2015*), language (*Fedorenko et al., 2010*) and music (*Fedorenko et al., 2011*), and even social tasks (e.g. *Deen et al., 2015*). Thus, the group difference for this area was expected given the behavioral results. The right insular cortex is multi-modal (*Bushara et al., 2003*), and is also involved in introspection (*Craig et al., 2000*). A comparison of  *Figures 1A and 1B* suggests that other regions might have mean group differences (e.g., frontal cortices), but due to large inter-subject variability in these regions, the group differences were not significant. This large

variability might account for the spurious dots of large τ values scattered throughout the cortical map (*Figures 1A and 1B*).

In addition to the whole-brain, data-driven analysis, we conducted an ROI, hypothesis-driven analysis. Given our previous ERP findings that P2 (and to a lesser extent N1) of individuals with dyslexia shows shorter adaptation (*Jaffe-Dax et al., 2017*), we conducted an ROI for the estimated cortical source of this ERP component. In a seminal MEG study (*Lütkenhöner and Steinsträter, 1998*), the source of N1 was attributed to Planum Temporale, and that of P2 to Heschl's gyrus. A more recent study (*Mayhew et al., 2010*) combined ERP and fMRI, and found significant correlations between variability in ERP-measured N1–P2 complex and BOLD responses in several regions, including the primary (Heschl's gyrus) and secondary (Planum Temporale and STG areas) auditory areas. Given these estimates, we conducted an ROI analysis on both primary and secondary auditory areas. We used a combined cytoarchitectonic (*Morosan et al., 2001*) and myeloarchitectonic (*Dick et al., 2012*) definition of these areas (the primary auditory cortex being composed of three sub-regions and two secondary auditory areas [Planum Temporale and Planum Polare]). We fitted the exponential decay model to the $\beta$s averaged over the right and the left primary auditory cortices (composed of 99 voxels each, denoted by the orange outlines in *Figure 1A–1B*), and over the two right and left secondary cortices. We found significant differences between the groups' τs in the left primary auditory cortex ($z$ = 2.6, p<0.01, effect size $r$ = 0.42; Mann-Whitney U-tests). In the right primary auditory cortex, the τ group difference showed the same trend, but did not reach significance ($z$ = 1.5, p=0.15, effect size $r$ = 0.23; Mann-Whitney U-tests). *Figure 2* shows the $\beta$s estimated for the left and right primary auditory cortices of the control (blue) and dyslexic (red) participants on each of the four TOA blocks. None of the other sub-regions of auditory cortex yielded significant group differences.

Taken together, the whole-brain and ROI analyses revealed a significant group difference in the timescales of adaptation in the left superior temporal cortex, left primary auditory cortex, and the right insular cortex. BOLD activity in both auditory regions was previously shown to correlate with the magnitude of N1–P2 responses (*Mayhew et al., 2010*). The right insular cortex is probably not associated with the P2 response that we measured with ERP and was therefore not predicted by our previous study. In addition, the general (though not significant) trend of shorter adaptation in participants with dyslexia was consistent across all responding voxels.

## Discussion

We characterized the cortical distribution of the decay of BOLD adaptation for participants with dyslexia and control participants, thus extending our previous behavioral and ERP study (*Jaffe-Dax et al., 2017*). We found a broadly distributed tendency for shorter adaptation in dyslexia. We further assessed group difference in the primary and secondary auditory cortices, associated with the production of P2 (*Mayhew et al., 2010*). Reports of previous studies asking whether these areas manifest neuro-typical structure and function in dyslexia are mixed. For example, *Clark et al. (2014)* reported early anatomical abnormalities, whereas *Boets et al. (2013)* reported adequate stimulus resolution. We now found a significant group difference in the left primary auditory cortex, and a similar tendency, which did not reach significance, in the right primary auditory cortex.

The broad distribution of dyslexics' faster decay of adaptation is in line with recent observations of a domain general abnormally small adaptation in dyslexia (*Perrachione et al., 2016*). The researchers compared BOLD responses to stimulus repetitions (in blocks of ~10 s) with BOLD response to non-repeated stimuli (auditory and visual), and found reduced stimulus-specific adaptation in the auditory (superior temporal), visual (fusiform and lateral occipital [LO]), and associative (insular and inferior frontal) cortices. Importantly, they found a three-way interaction of group (control and dyslexic) x condition (repeated and non-repeated) x time (within the block), where controls' increase of adaptation along the block was larger than that of participants with dyslexia. This observation, which results from controls' accumulative adaptation with stimulus repetition, is fully consistent with our observation of dyslexics' faster decay of adaptation.

We should note that the timescale of impaired retention in dyslexia is slower than that of sensory memory (iconic or echoic memory is <<1 s). Still, it is expected to affect the perception of individuals with dyslexia owing to their reduced cross-trial retention. It is therefore expected to impede their performance in a broad range of perceptual tasks, such as entrainment to rapid stimuli (e.g.,

*Witton et al., 1998*; *Goswami, 2011*; *Lehongre et al., 2013*), but only in protocols that contain stimulus repetitions. Indeed, it was shown that listeners entrain better to familiar compared with unfamiliar stimuli (*Doelling and Poeppel, 2015*; *Kumagai et al., 2017*). The advantage of familiarity is expected to be smaller in dyslexia. The faster decay of implicit perceptual memory of individuals with dyslexia is expected to also impede their long-term accumulation of stimulus statistics, and consequently to reduce the complexity and richness of their long-term accumulated categorical representations (e.g., *Perrachione et al., 2011*; *Banai and Ahissar, 2017*).

In summary, the data collected in this study point to the specific neural structures that underlie the 'anchoring deficit' in dyslexia, namely a reduced use of stimulus statistics (*Ahissar et al., 2006*; *Ahissar, 2007*; *Oganian and Ahissar, 2012*). These data suggest a broad multimodal (e.g., *Jaffe-Dax et al., 2016*) cortical distribution that includes, but is not limited to, sensory areas.

## Materials and methods

In the two-tone frequency discrimination task, subjects were asked to indicate which of two sequentially presented tones had a higher pitch. The tones were 50 ms long, presented at comfortable intensity, and were drawn from a uniform distribution between 800 Hz and 1250 Hz. The frequency difference within each pair was randomly drawn at between 1% and 20% (following the protocol in *Jaffe-Dax et al., 2017*). In the pre-training session (8 min), each participant performed one block of each of the four Trial Onset Asynchronies (TOAs) of 3, 6, 9, or 15 s, administered in four separate blocks in random order (each block consisted of 16 trials). These TOAs are longer than those in our previous ERP experiment (1.5, 3, 6, and 9 s, *Jaffe-Dax et al., 2017*), because the controls' ERP (N1 and P2) response at 9 s was still larger than that at 6 s. Each block had a constant TOA of 3, 6, 9, or 15 s. In the scanner, each participant performed three runs of four blocks (of 16 trials). The block design allowed us to measure the dynamics of adaptation in timescales that are independent of the sluggishness of typical hemodynamic response function (HRF), as we analyzed each block as a whole, and not on a trial-by-trial basis. Specifically, we modelled the magnitude of the BOLD signal in each block as a function of its TOA. This block design was used to estimate $\tau$ on the basis of the

**Table 1.** General characteristics of the participants in this study (mean and standard deviation). The assessments used in this study were the same as in our previous study (*Jaffe-Dax et al., 2017*).

|  | Control group | Dyslexic group | Mann-Whitney z value |
|---|---|---|---|
|  | N = 19 | N = 20 |  |
| Age (years) | 25.9 (2.6) | 24.5 (2.6) | 1.7 n.s. |
| General cognitive (scaled) |  |  |  |
| Block design | 13.1 (3.2) | 12.4 (2.9) | 0.6 n.s. |
| Digit span | 11.1 (2.9) | 7.5 (1.9) | 4.0**** |
| Phonological speed [items/minute] |  |  |  |
| Pseudo-word reading rate | 64.0 (25.2) | 31.9 (9.7) | 3.9**** |
| Single-word reading rate | 101.6 (35.2) | 69.2 (21.3) | 3.0*** |
| Word pattern recognition rate | 69.8 (15.6) | 41.7 (11.8) | 4.5**** |
| Passage reading rate | 142.2 (23.9) | 100.7 (17.4) | 4.5**** |
| Spoonerism rate | 9.9 (3.0) | 5.7 (3.1) | 3.8**** |
| Phonological accuracy [% correct] |  |  |  |
| Pseudo-word reading accuracy | 90.6 (11.9) | 63.5 (18.4) | 4.0**** |
| Single-word reading accuracy | 97.2 (4.3) | 89.0 (6.5) | 3.7**** |
| Word pattern recognition accuracy | 100.0 (0.0) | 98.27 (3.1) | 2.5** |
| Passage reading accuracy | 98.7 (1.2) | 95.4 (2.3) | 4.1**** |
| Spoonerism accuracy | 90.8 (6.9) | 77.8 (17.2) | 2.5** |

*p < 0.05; **p<0.01; ***p<0.005; ****p<0.0005.
DOI: https://doi.org/10.7554/eLife.30018.004

magnitude of the BOLD response. However, the number of trials was too small for robust estimation of behavioral context effects, which are based on the difference in success rate (binary scores for each trial) between trials that gain and those that are hampered by the context (*Jaffe-Dax et al., 2017*). Stimuli were digitally constructed using Matlab 2015b (The Mathworks Inc., Natwick, MA, USA) and administered through inserted sound-attenuating MR compatible S14 earphones (Sensimetrics Corporation, Malden, MA, USA). The demographic, cognitive and reading assessments of this cohort are described in *Jaffe-Dax et al. (2017)*.

Before the functional scan, high-resolution (1 × 1 × 1 mm resolution) T1-weighted magnetization-prepared rapid acquisition gradient-echo (MPRAGE) images were acquired using a 3T Magnetom Skyra Siemens scanner and a 32-channel head coil at the ELSC Neuroimaging Unit (ENU). The cortical surface was reconstructed from the high-resolution anatomical images using standard procedures implemented by the BrainVoyager QX software package (version 2.84; Brain Innovation, The Netherlands). The functional T2*-weighted MRI protocols were based on a multislice gradient echo-planar imaging and obtained under the following parameters: TR = 1 s, TE = 30 ms, flip angle = 90°, imaging matrix = 64 × 64, field-of-view=192 mm; 42 slices with 3 mm slice thickness and no gap were oriented in AC-PC plane, covering the whole brain, with functional voxels of 3 × 3 × 3 mm and multiband parallel imaging with an acceleration factor of 3 (*Moeller et al., 2010*).

Preprocessing of functional scans in BrainVoyager included 3D motion correction, slice scan time correction, and removal of low frequencies up to three cycles per scan (linear trend removal and high-pass filtering). The anatomical and functional images were transformed to the Talairach coordinate system using trilinear interpolation. Each voxel's time course was z-score normalized and smoothed using a 3D Gaussian filter (full width at half maximum [FWHM] of 4 mm). A standard (two gamma) hemodynamic response function (*Friston et al., 1998*) was convolved with the trial timings of each TOA block to build four predictors for the subsequent GLM analysis. For all task-responsive voxels ($p<0.001$, FDR corrected; *Benjamini and Yekutieli, 2001*), each TOA condition was modeled separately to account for its contribution to the measured BOLD signal in each voxel. Specifically, a single $\beta$ value was obtained for each TOA condition. An exponential decay model (see 'Results') was fitted to these $\beta$ values, and its parameters were estimated for each voxel in each subject using a least-square method. For ROI analysis, the MNI coordinates of auditory cortex subdivision were obtained from *Morosan et al., 2001* and translated into Talairach coordinates using Yale BioImage Suite Package (sprout022.sprout.yale.edu/mni2tal/mni2tal.html; *Lacadie et al., 2008*). The BOLD signal was averaged for each ROI and then the $\beta$ values of the four TOA blocks were fitted to the exponential decay.

Whole-brain significance results were corrected for multiple comparison false-positive biases by a Monte-Carlo cluster correction (*Forman et al., 1995*), implemented using a plug-in of BrainVoyager (*Goebel et al., 2006*). The Monte-Carlo procedure was given an a-priori chosen probability for type I error of 0.05 and yielded a cluster threshold of 44 voxels. Non-parametric tests (Mann-Whitney's U-test) were used for group comparisons, as we did not assume a normal distribution, as in our previous study (*Jaffe-Dax et al., 2017*).

## Acknowledgements

We thank Udi Zohary, Yuval Porat, Luba Daikhin, Tal Golan and Zvi Roth for their valuable feedback on this manuscript.

---

## Additional information

### Funding

| Funder | Grant reference number | Author |
| --- | --- | --- |
| Gatsby Charitable Foundation | | Merav Ahissar |
| German-Israeli Foundation for Scientific Research and Development | I-1303-105.4/2015 | Merav Ahissar |
| Israel Science Foundation | 1650/17 | Merav Ahissar |

Canadian Institute for Advanced Research                                          Merav Ahissar

The funders had no role in study design, data collection and interpretation, or the decision to submit the work for publication.

## Author contributions

Sagi Jaffe-Dax, Conceptualization, Data curation, Software, Formal analysis, Supervision, Validation, Investigation, Visualization, Methodology, Writing—original draft, Project administration, Writing—review and editing; Eva Kimel, Investigation, Data curation, Writing—original draft, Writing—review and editing; Merav Ahissar, Conceptualization, Supervision, Funding acquisition, Writing—original draft, Project administration, Writing—review and editing

## Author ORCIDs

Sagi Jaffe-Dax  http://orcid.org/0000-0002-8759-6980

## Ethics

Human subjects: Informed consent was acquired from all participants. The study was approved by The Hebrew University Committee for the Use of Human Subject in Research.

## Decision letter and Author response

Decision letter https://doi.org/10.7554/eLife.30018.008
Author response https://doi.org/10.7554/eLife.30018.009

## Additional files

### Supplementary files

• Transparent reporting form
DOI: https://doi.org/10.7554/eLife.30018.005

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
