## [Decision Letter]

Thank you for submitting your article "Widespread shorter cortical adaptation in dyslexia" for consideration by *eLife*. Your article has been reviewed by three peer reviewers, and the evaluation has been overseen by a Reviewing Editor and Andrew King as the Senior Editor. The following individual involved in review of your submission has agreed to reveal his identity: John Stein (Reviewer #2).

The reviewers have discussed the reviews with one another and the Reviewing Editor has drafted this decision to help you prepare a revised submission.

Summary:

This study builds on the authors' previous work that demonstrated using behavioral measurements and EEG recordings that dyslexics show a faster decay of implicit memory than controls, which may account for their longer reading times. Here they used fMRI to investigate which cortical areas show these shorter neural adaptation effects. The authors calculate time constants of adaptation and find these to be significantly different in the dyslexic subjects in a region of left non-primary cortex and in the right anterior insula close to frontal operculum. The reviewers agreed that this study has a solid theoretical and empirical foundation in the previous work carried out by the authors and by other groups, and that the results are presented clearly and potentially add important new details to the framework for understanding dyslexia. The success of the authors in demonstrating differences in the time constant between the groups using the sluggish BOLD response is notable. Nevertheless, the reviewers had some difficulty reconciling these data with the previous neurophysiological data.

Essential revisions:

1) The study does not show widespread significant shorter cortical adaptation as suggested in the title. The regions in which significant group differences are small and include high-level multimodal cortex that is not predicted by the previous work. The N1 and P2 responses are generally regarded as arising from non-primary auditory cortex in the posterior superior plane and the previous neurophysiological data fitted with the idea that there might be a less stable representation of sensory information over time in dyslexia promoted by this group, which is a compelling idea. A specific test of this idea would require a comparison of the time constants in the generator of the N1, conventionally regarded as in planum temporale (there are probably two generators in PT as suggested by the work of Lutkenhoner). The reviewers were unanimous that a region of interest analysis is needed, though this might be focused on this region for which there is a strong prior based on the previous work.

2) Examination of the areas in which there is a decrease in tau without correction for multiple comparisons shows a widespread swathe of differences that I find difficult to reconcile with the hypothesised deficit in the sensory trace. For example, there is shorter tau in primary somatosensory cortex and motor cortex. Do the authors think this is consistent with the anchoring deficit model?

3) The authors suggest that poorer behavioral performance by the dyslexia group in the scanner could be related to their impairment in learning the task after training compared to the control group. An alternative possibility that needs to be discussed is that these results are due to the added acoustic noise of scanning, since continuous fMRI sampling as opposed to sparse sampling was employed. It is well known that individuals with dyslexia tend to have trouble with noise exclusion, and a tone-discrimination task with relatively sensitive pitch differences will be disproportionately difficulty for dyslexic individuals in the presence of noise. There is a paper by Chait et al., 2007 to that effect (tones in noise) specifically, if a citation is needed, as well as the larger noise-exclusion literature (e.g. Sperling et al., 2005 and others).

4) Some attempt should be made to explain why the decay is faster in dyslexics by fitting the results into what is already known about dyslexics' relative insensitivity to amplitude and frequency modulations. For example, lower frequency sensitivity may impair auditory entrainment to a tone; could this underlie the difference in adaptation time constant?

5) All the reviewers were struck by the capacity to extract adaptation and adaptation-recovery time course information from fMRI data, but it was agreed that a clearer explanation is warranted of how this can be studied using a method that has a time lag of several seconds.

6 The characteristics of the dyslexics should be given in brief for those who haven't read the previous paper. More details are also needed regarding the Monte Carlo simulations for cluster-level multiple comparison corrections to ensure that others could replicate this approach.

7) The discussion of the time course of adaptation with regards to the Perrachione et al.,2016 at the beginning of the Discussion section is confusing. In that paper, stimuli were presented for 8 repetitions in blocks of approximately 10 seconds, but each adapting or non-adapting stimulus was presented for 700 ms and at a rate of one per 1200 ms - so when the authors say "adaptation across a window of ~10 seconds was smaller than controls because it was largely recovered" it is unclear how they reconcile this with the stimulation procedure of the Perrachione study.

8) It should also be pointed out that Perrachione et al., 2016 reported a group x time x condition interaction in the adaptation response in their study (see Supplementary table S4) for all conditions, which they interpreted as evidence for increasing adaptation in the control group w/ repeated presentation of the same stimulus, whereas such increasing adaptation was attenuated or absent in the dyslexia group. The correspondence between this result and the present observations seems meaningful and is worth commenting on.

---

## [Author Response]

Essential revisions:1) The study does not show widespread significant shorter cortical adaptation as suggested in the title. The regions in which significant group differences are small and include high-level multimodal cortex that is not predicted by the previous work. The N1 and P2 responses are generally regarded as arising from non-primary auditory cortex in the posterior superior plane and the previous neurophysiological data fitted with the idea that there might be a less stable representation of sensory information over time in dyslexia promoted by this group, which is a compelling idea. A specific test of this idea would require a comparison of the time constants in the generator of the N1, conventionally regarded as in planum temporale (there are probably two generators in PT as suggested by the work of Luktenhoner). The reviewers were unanimous that a region of interest analysis is needed, though this might be focused on this region for which there is a strong prior based on the previous work.

Thank you for this comment.

1) We changed the title of the paper to "Shorter cortical adaptation in dyslexia is broadly distributed in the superior temporal lobe and includes the primary auditory cortex".

2) The description of our ROI analysis now makes this point explicit. We now clarify that our ROI analysis was guided by the literature that mapped the cortical sources of P2 and N1 (Lütkenhöner and Steinsträter, 1998; Mayhew et al., 2010). The estimated source is somewhat broad, and includes all auditory areas, primary and secondary (PT and STG). We originally conducted an ROI mask for the primary regions (3 sub-regions, as in the original manuscript). We now added secondary auditory (PT and Planum Polaris) regions. Only one left primary sub-region showed significant group effect. Our whole brain STG observation is also broadly in line with Mayhew et al., (who found that STG is also correlated with variability in N1 and P2), yet specific coordinates are not provided in their paper.

2) Examination of the areas in which there is a decrease in tau without correction for multiple comparisons shows a widespread swathe of differences that I find difficult to reconcile with the hypothesised deficit in the sensory trace. For example, there is shorter tau in primary somatosensory cortex and motor cortex. Do the authors think this is consistent with the anchoring deficit model?

Yes. We added a short paragraph to the Discussion section to clarify this point. Behavioral observations (e.g., Jaffe-Dax et al., 2016) and imaging data (e.g., Perrachione et al., 2016) are consistent with a multimodal anchoring deficit that has specific temporal characteristics. This faster decay is "post sensory" (i.e. duration of echoic and iconic memories <1 second) and yet it affects perception. The study shows that poor cross trial retention is manifested in poor retention in primary and higher-level cortical areas.

3) The authors suggest that poorer behavioral performance by the dyslexia group in the scanner could be related to their impairment in learning the task after training compared to the control group. An alternative possibility that needs to be discussed is that these results are due to the added acoustic noise of scanning, since continuous fMRI sampling as opposed to sparse sampling was employed. It is well known that individuals with dyslexia tend to have trouble with noise exclusion, and a tone-discrimination task with relatively sensitive pitch differences will be disproportionately difficulty for dyslexic individuals in the presence of noise. There is a paper by Chaitet al., 2007 to that effect (tones in noise) specifically, if a citation is needed, as well as the larger noise-exclusion literature (e.g. Sperling et al., 2005 and others).

Thank you for pointing this out. We now clarify that the training period was conducted with noise simulating the noise in the scanner, which explains why performance of both groups was not reduced in the scanner compared to their training performance.

4) Some attempt should be made to explain why the decay is faster in dyslexics by fitting the results into what is already known about dyslexics' relative insensitivity to amplitude and frequency modulations. For example, lower frequency sensitivity may impair auditory entrainment to a tone; could this underlie the difference in adaptation time constant?

This is an important point. We predict that dyslexics' entrainment to unfamiliar stimuli will not be impaired. However, dyslexics' benefits from exposure are reduced compared with good reading controls. Since repetition lowers the measured thresholds, and since most protocols contain stimulus repetitions, we expect dyslexics to have poorer thresholds in a broad range of perceptual tasks (e.g., Witton et al., 1998; Goswami, 2011; Lehongre et al., 2013). For entrainment, it was specifically shown that tracking familiar stimuli is better than tracking similar yet unfamiliar stimuli. We now explain this in the Discussion section.

5) All the reviewers were struck by the capacity to extract adaptation and adaptation-recovery time course information from fMRI data, but it was agreed that a clearer explanation is warranted of how this can be studied using a method that has a time lag of several seconds.

The time lag of the fMRI signal is not detrimental to the estimation of the τ values more than to the standard β estimation. In this study we used a block design; each block was defined by its Trial Onset Asynchrony (TOA), and for each TOA there were 3 blocks. Then, a standard modeling with 4 conditions was applied to extract β values for each TOA condition (each calculated from a different block). Next, a model for estimating the decay time constant (τ) value was applied to each voxel: βTOA=a+bexp-TOA/τ, where τ denotes the timescale of adaptation, a is the asymptote level of the BOLD signal and b is the amplitude of adaptation. We added clarifications in the Materials and methods section regarding the nature and benefits of this block design.

6 The characteristics of the dyslexics should be given in brief for those who haven't read the previous paper.

Group characteristics were now added to the paper (new Table 1).

More details are also needed regarding the Monte Carlo simulations for cluster-level multiple comparison corrections to ensure that others could replicate this approach.

The cluster correction was performed using the cluster-size thresholding plugin of the BrainVoyager QX software (version 2.84; Brain Innovation, The Netherlands). The size of the cluster is determined by the significance level that we decide a-priori, as now better clarified in the text (a detailed description of this correction method is provided in Goebel et al., 2006, who implemented it in BrainVoyager).

7) The discussion of the time course of adaptation with regards to the Perrachione et al., 2016 at the beginning of the Discussion section is confusing. In that paper, stimuli were presented for 8 repetitions in blocks of approximately 10 seconds, but each adapting or non-adapting stimulus was presented for 700 ms and at a rate of one per 1200 ms - so when the authors say "adaptation across a window of ~10 seconds was smaller than controls because it was largely recovered" it is unclear how they reconcile this with the stimulation procedure of the Perrachione study.

Note that we refer to duration of adaptation as the time adaptation lasts rather than the duration of its induction. The duration of adaptation is characterized by the magnitude of the response within the time window of the repetition Block. Perrachione et al., 2016 tracked responses over 10 second (Block duration), and hence tracked the accumulative impact of adaptation as a function of repetition within this time window. The magnitude and duration of (the induced) adaptation both increase with repetition, but to a lesser extent in dyslexics. We now explain this more clearly in the Discussion section.

8) It should also be pointed out that Perrachione et al., 2016 reported a group x time x condition interaction in the adaptation response in their study (see Supplementary table S4) for all conditions, which they interpreted as evidence for increasing adaptation in the control group w/ repeated presentation of the same stimulus, whereas such increasing adaptation was attenuated or absent in the dyslexia group. The correspondence between this result and the present observations seems meaningful and is worth commenting on.

Thank you for pointing this out. We added this to the Discussion section – this is exactly our prediction – that controls' adaptation is accumulated across repetitions to a larger extent than dyslexics', due to the faster decay of dyslexics' implicit memory trace within this time window (~10 second, see comment 8). We now clarify it in the manuscript.